# Functional Interactions between Entorhinal Cortical Pathways Modulate Theta Activity in the Hippocampus

**DOI:** 10.3390/biology10080692

**Published:** 2021-07-21

**Authors:** Víctor J. López-Madrona, Santiago Canals

**Affiliations:** 1Institut de Neurosciences des Systèmes, Aix Marseille University, INSERM, INS, 13005 Marseille, France; 2Instituto de Neurociencias, Consejo Superior de Investigaciones Científicas and Universidad Miguel Hernández, Sant Joan d’Alacant, 03550 Alicante, Spain

**Keywords:** hippocampus, entorhinal cortex, theta, Granger causality, information processing, connectivity, independent component analysis

## Abstract

**Simple Summary:**

The activity in the hippocampus is characterized by a strong oscillation at theta frequency that organizes the neuronal firing. We have recently shown that different theta oscillations are present in the hippocampus, opening the possibility to multiple interactions between theta rhythms. In this work, we analyzed the functional connectivity between theta generators during the exploration of a known environment with or without a novel stimulus. The directionality of the interactions was determined using tools based on Granger causality and transfer entropy. We found significant interactions between activity components originated in CA3 and in layers II and III of the entorhinal cortex. During exploration with a novel stimulus, the connectivity from the entorhinal cortex layer II increased, while the influence of CA3 decreased. These results suggest that the entorhinal cortex layer II may drive theta interactions and synchronization in the hippocampus during novelty exploration.

**Abstract:**

Theta oscillations organize neuronal firing in the hippocampus during context exploration and memory formation. Recently, we have shown that multiple theta rhythms coexist in the hippocampus, reflecting the activity in their afferent regions in CA3 (Schaffer collaterals) and the entorhinal cortex layers II (EC-II, perforant pathway) and III (EC-III, temporoammonic pathway). Frequency and phase coupling between theta rhythms were modulated by the behavioral state, with synchronized theta rhythmicity preferentially occurring in tasks involving memory updating. However, information transmission between theta generators was not investigated. Here, we used source separation techniques to disentangle the current generators recorded in the hippocampus of rats exploring a known environment with or without a novel stimulus. We applied analytical tools based on Granger causality and transfer entropy to investigate linear and non-linear directed interactions, respectively, between the theta activities. Exploration in the novelty condition was associated with increased theta power in the generators with EC origin. We found a significant directed interaction from the Schaffer input over the EC-III input in CA1, and a bidirectional interaction between the inputs in the hippocampus originating in the EC, likely reflecting the connection between layers II and III. During novelty exploration, the influence of the EC-II over the EC-III generator increased, while the Schaffer influence decreased. These results associate the increase in hippocampal theta activity and synchrony during novelty exploration with an increase in the directed functional connectivity from EC-II to EC-III.

## 1. Introduction

The electrophysiological activity of the hippocampus is characterized by the presence of strong oscillatory activity in the theta (4–8 Hz) and gamma (30–100 Hz) frequency bands [1]. Theta rhythmicity is recorded across all hippocampal sub-regions and shows its largest amplitude in the str. lacunosum-moleculare of CA1 [1]. This activity results from the interaction of several theta current source generators entrained by multiple theta rhythm oscillators, including the medial septum [2], nucleus incertus [3], subiculum [4], and the entorhinal cortex (EC) [5,6]. Therefore, although it is commonly recorded as a unique oscillation, multiple and relatively independent theta activities coexist in the hippocampus [6,7]. Similarly, distinct gamma oscillations are found in the different hippocampal regions [8,9], which can be nested to the different theta oscillations [7]. For instance, gamma activity in CA1 is characterized by slow (~30–60 Hz) and medium (~60–100 Hz) frequency bands that are coherent with neuronal firing and gamma activities in the afferent regions in CA3 and EC, respectively [10,11], although further subdivisions of the gamma band have been recently proposed [12,13]. Several gamma bands have been also reported in the dentate gyrus (DG) [14,15], including a slow (~30–50 Hz) and a fast gamma activity (~100–150 Hz) also nested to the local theta rhythm [7,16]. The former is synchronous with the lateral EC and it has been related to object learning (“what” pathway), while the latter is coherent with the medial EC and would encode spatial learning (“where” pathway) [16].

The interaction between hippocampal activities at different frequency bands has been proposed to play an important role in memory formation [7,15,16,17]. For instance, inputs from CA3 and EC to CA1 organized in gamma oscillations but segregated in different phases of the same theta cycle, have been associated to memory retrieval and encoding, respectively, as a mechanism to differentiate computations [18,19,20]. Similarly, distinct gamma frequencies nested to the theta oscillation have been proposed as a mechanism to multiplex information and further segregate information channels in the hippocampus [10]. In the context of the new findings highlighting multiple theta activities in the hippocampus [7], we have proposed a mechanism based on theta-gamma cross-frequency coupling to modulate the synchrony of region-specific theta oscillations, thus regulating information transmission associated to the phase of the theta cycle. Consistent with the idea that the retrieval of stored memories is supported by the CA3 network and stimuli encountered in the environment conveyed by the EC inputs, we found increased theta-gamma cross-frequency coupling and theta synchrony between CA3 and EC-associated generators when animals explored novel environments and updated the existing memory with new information [7]. However, a formal analysis of information transmission was not performed.

Granger Causality (GC) applied to electrophysiological signals allows the investigation of the ‘causal’ influence of one brain area over another [21,22,23,24]. The concept of causality in GC is based on prediction, where the past activity of the sender can help to predict the future dynamics of the receiver, and it is implemented in practice using autoregressive models. Numerous extensions and variants of GC have been developed, either overcoming some of its limitations or incorporating new information and interpretations to the results. Guo and colleagues [25] proposed a partial GC, which estimates the GC after removing the influence of unknown variables (i.e., time-series that were not included in the analysis). It can be estimated whether the receiver follows directly or inversely the activity of the sender, what was applied in computational models to distinguish two circuits with different mechanisms (excitation and inhibition) but identical GC values [26]. Baccalá and Sameshima introduced a new approach, termed Partial Directed Coherence (PDC) [27], to estimate a frequency-domain representation of the direct GC between two nodes, i.e., controlling the influence of other variables included in the model. However, a common limitation to all these approaches is that they are restricted to linear interactions. Alternatively, methodologies based on Information Theory, as Transfer Entropy (TE) [28], estimate the connectivity using conditional entropies, allowing the identification of linear and nonlinear interactions, with the counterpart that they require larger datasets and cannot provide a spectral decomposition. It has been recently proposed that the combination of both frameworks (GC and Information Theory) is used to estimate the connectivity in short data segments and without assuming linearity [29]. 

The interpretation of directed interdependencies between neuronal signals in the hippocampal circuit is, however, not straightforward. The polysynaptic loops between the EC and the hippocampus make that variations in connectivity strength between EC layers completely alter the distribution of functional links between hippocampal regions [30]. In the present work, we used blind separation techniques, applied to high density electrophysiological recordings, to separate two distinct EC activities originating in EC-II and EC-III, respectively, and one corresponding to the Schaffer collateral afference in CA1 to investigate the directed functional connectivity between these pathways. Our results link increased hippocampal theta activity and synchrony during novelty exploration to an increase in the information flow from EC-II to EC-III.

## 2. Materials and Methods

### 2.1. Animals and Surgery

Five adult male Long-Evans rats (250–300 g) were each implanted with a 32 channels silicon probe (Neuronexus Technologies, MI, USA) across the dorsal hippocampus (data are available at http://dx.doi.org/10.20350/digitalCSIC/12537 (accessed on 19 July 2021)). Data from the same subjects have been used in a previous study [7]. An Ag/AgCl wire (World Precision Instruments, Sarasota, FL, USA) electrode was placed in contact with the skin on the sides of the surgery area and used as ground. The data were acquired at 5 kHz, with an analog high-pass filter at 0.5 Hz. After digitalization, we low-pass filtered the signals at 300 Hz, removed the line noise at 50 Hz and its first harmonic with Notch filters and down-sampled the signals at 2.5 kHz. At the end of the experiments, animals were perfused with 4% paraformaldehyde, and the final position of the electrodes confirmed histologically.

The rats were left for at least 10 days after the surgery, until they recovered completely. During the first 72 h, animals were injected subcutaneously with analgesic twice per day (Buprenorphine, dose 2–5 μg/kg, RB Pharmaceutical Ltd., Berkshire, UK). Antibiotic (Enrofloxacin, 10 mg/kg, Syva, León, Spain) dissolved in the drinking water was also provided during the first post-surgery week. Behavioral training did not start until the animals showed no signs of discomfort with the manipulation of the implants.

### 2.2. Data Acquisition

We carried out a test of mismatch novelty in all subjects. First, we did a habituation process with two sessions per day during 11 days before the surgery and 8 days after recovery. Each session consisted in 10 min freely exploring an open field (plexiglass sandbox of 50 × 50 cm, opened at the top and with three visual cues in three of the walls). The 9th day after recovery, we performed the “novelty” test. For 10 min, the subjects were introduced in a “novelty chamber” consisting of a transparent methacrylate box inside the familiar environment, with a square base 35 cm wide and 40 cm high, and with sandpaper on the floor to provide a noticeable tactile stimulus. This corresponded to the “novelty session”. Immediately after that, the animals were left in the original open field for another 10 min, which was considered as the control condition. 

Local field potentials (LFP) were acquired at 5 kHz, with an analog high-pass filter at 0.5 Hz. After digitalization, signals were low pass filtered at 300 Hz and the net noise and its first harmonic were removed with Notch filters at 50 and 100 Hz. Data was down sampled at 2.5 kHz.

### 2.3. Independent Component Analysis

An extensive description of the methodology applied on the present dataset can be found in [7]. Briefly, the aim of independent component analysis (ICA) is to separate N statistically independent sources that are mixed on M sensors (where N ≤ M). As the distributions of the sources are unknown, it performs a blind separation of patterns. Moreover, it assumes that the sources have different spatial distributions, they are spatially immobile, and their time-courses are independent, although it is robust to even high levels of source correlation [31]. Each recorded LFP *u_m_(t)* is then modeled as the sum of *N* sources *s_n_(t)* multiplied by a constant weight factor *V_mn_*:(1)$$u_{m}\left(t\right)=\overset{N}{\underset{n=1}{\sum }}V_{mn}s_{n}\left(t\right),\;\;m=1,2,\ldots ,M$$
where *V_mn_* is the mixing matrix or topography and represents the voltage contribution of the source *n* to the sensor *m*.

ICA has been long used in surface recordings, removing noisy components, such as cardiac activity or eye blinking [32], and identifying task-related neural sources [33,34] or epileptic generators [35,36]. Moreover, it has been proposed as a tool to disentangle deep brain activities hidden at the surface by superficial signals with higher amplitudes [37]. Although its use in intracranial recordings is reduced in comparison, its effectiveness has been well studied [31,38], and it has been used to dissociate local and remote cortical field potentials [39,40] and different current generators in the hippocampus [7,16,41,42]. 

A main difficulty when computing ICA is the correct identification of physiological sources. As ICA may identify as many possible sources as the number of LFP signals, additional constraints are necessary to ensure the origin, either neuronal or noisy, of each generator. First, the anatomical distribution of each current generator is assumed to be fixed, thus the voltage distribution or topography of the generator should be stable over time. This can be tested by fragmenting the signal into smaller time windows and computing ICA on each fragment separately. Moreover, a certain degree of similarity between generators is expected in different subjects with similar electrode implantations. Second, the topography should follow the specific distribution of axons and dendrites composing the source. This requires a previous knowledge of the anatomical substrate and realistic computational models to simulate the currents. Third, to ensure the synaptic specificity of each generator, the different substructures can be independently modulated, for example, with pharmacology, electrical or optogenetic stimulation, confirming that only the associated generator is affected by the manipulation. 

We have applied ICA on the continuous data for each condition, control, and novelty, extracting three common and stable components (named as independent components of the local field potentials, IC-LFPs) in all subjects that correspond to pathway-specific inputs to the hippocampus. Two of them were in CA1, one in stratum radiatum, which corresponds to the synaptic terminals of Schaffer collaterals from CA3 to the pyramidal cells in CA1 (Sch-IC; *IC* referring to independent component) [41,43,44,45,46,47]. The other component has a current sink in stratum lacunosum-moleculare (lm-IC), where the inputs from EC-III to the pyramidal cells of CA1 are located [41,43,47]. A third component was identified in the DG, which corresponds to the axons projected from EC-II to the dendrites of the granular cells through the perforant pathway (PP-IC) [7,16,43,45,46]. 

In this work, ICA was computed using the algorithm RUNICA, which performs the separation based on maximal entropy [48]. It is implemented in the matlab toolbox ‘ICAofLFPs’, available at http://www.mat.ucm.es/~vmakarov/downloads.php accessed on 19 July 2021.

### 2.4. Power Analysis of Time Series

As ICA does not ensure the correct polarity and amplitude of each generator [49], we normalized the power of the IC-LFPs, imposing to each dataset (i.e., animal) an averaged mean value of zero and a standard deviation equal to the unit. This way, we increased the similarities inter-subject and facilitated their comparison. We used the multitaper method [50] to compute the power spectra of the IC-LFPs. We selected all the continuous data for each session. Based on previous studies [7] we defined the following frequency bands: theta (6–10 Hz), slow gamma (30–60 Hz), medium gamma (60–100 Hz), and fast gamma (100–150 Hz). The power of each band was computed as the mean value of the power spectrum at that frequency band.

### 2.5. Partial Directed Coherence

To assess the directionality of the connectivity between IC-LFPs we used a PDC analysis [27]. PDC is based on autoregressive models, same as the well-known GC [21]. Given two time series *x(t)* and *y(t)*, it considers that there is a directed functional connectivity from *y(t)* to *x(t)* if the past of *y(t)* can be used to predict the future values of *x(t)*. The value of *x(t)* at each time point is modeled as the summation of the previous *p* values of *x(t)* (where *p* represents the model order) plus the previous *p* values of *y(t)*. If the error of the prediction combining the past of *x(t)* and *y(t)* is lower than using the past of *x(t)* alone, then it is considered that *y(t)* ‘granger-causes’ *x(t)*. 

PDC performs the Fourier Transform on the autoregressive model to obtain a spectral decomposition of the coefficients. Thus, it characterizes the directionality at each frequency band. PDC identifies the direct interaction between signals, excluding the contribution from indirect pathways (for example, mediated by a third variable). Finally, PDC normalizes the connectivity between 0 and 1 by computing the ratio between the flow from the sender to the receiver divided by all the outflows from the sender.

Before computing PDC, the IC-LFPs were low-pass filtered at 100 Hz and down sampled at 250 Hz. We computed the PDC on the continuous data for each condition with a sliding window of 5 s without overlap, averaging the results across windows. We selected a model order of 20 samples (80 milliseconds). We varied the model order between 15 and 25 samples, with similar results in all cases. We computed PDC using the extended multivariate autoregressive modelling (eMVAR) toolbox [51].

### 2.6. Partial Transfer Entropy

To further extend the directionality analysis, we computed Partial Transfer Entropy (PTE) between the IC-LFPs [28,52]. In contrast with PDC, which computes linear autoregressive models, PTE is based on the information theory framework and it determines that there is a connectivity from one signal to another if the conditional entropy of the latter is reduced when the past of the former is included. PTE estimates both linear and nonlinear interactions at the expense of being a temporal index (i.e., it has no spectral decomposition). Therefore, PTE and PDC can be used as complementary approaches.

PTE is an extension of the transfer entropy (TE) to differentiate only the direct interactions between the signals. Given two time-series *X(t)* and *Y(t)*, TE is defined as:

(2)$$TE_{X\to Y}=H\left(Y_{t}|Y_{t-1:t-\tau }\right)-H(Y_{t}|Y_{t-1:t-\tau },X_{t-1:t-\tau })$$where *H* denotes the entropy and τ is a temporal delay. TE measures the reduction in entropy on the state of one variable (*Y(t)*) when the past of the second one (*X(t)*) is included alongside the past of the former. Analogously to PDC, the PTE is a partialized version of the TE, including additional variables in the conditional entropies:

(3)$$PTE_{X\to Y,Z}=H\left(Y_{t}|Y_{t-1:t-\tau },Z_{t-1:t-\tau }\right)-H(Y_{t}|Y_{t-1:t-\tau },Z_{t-1:t-\tau },X_{t-1:t-\tau })$$where $Z=\left\{Z_{1,t},Z_{2,t},\ldots ,Z_{k,t}\right\}$ is a multivariate set of *k* random variables. PTE measures the influence of the past of *X(t)* on the present of *Y(t)*, which cannot be accounted for the other *k* variables.

We followed the same preprocessing as for PDC analysis. The IC-LFPs were low-pass filtered at 100 Hz and down sampled at 250 Hz, and PTE was estimated using a sliding window of 5 s without overlap. PTE was computed using the MuTE toolbox [53], which is also available inside the HERMES framework [54].

### 2.7. Statistical Analysis

To compare the power spectra between conditions, we used a two-way ANOVA at each frequency band (theta, slow-gamma, medium-gamma and fast-gamma). For each ANOVA, we performed a paired comparison between control and novelty conditions, including each IC-LFP as a factor. We used Holm-Šídák to correct for multiple comparisons. A similar procedure was used in the PDC for the theta frequency band but including each pair of IC-LFPs connections as a factor. To test whether the temporal dynamics of the PDC and PTE change in time during the exploration trial, we computed the correlation between the connectivity values through all time windows and the duration of the task for each subject. Then, we tested if the correlation values across subjects where different from zero using a t-test. Thus, we tested whether the PDC and PTE increased or decreased linearly during the trial. The differences in the PTE analysis were computed using an uncorrected paired t-test between the PTE values of each pair of IC-LFPs in control versus the novelty condition.

## 3. Results

### 3.1. Theta and Gamma Oscillations Are Enhanced during Novelty Exploration

Electrophysiological recordings were performed while the animals freely explored a familiar (control condition) or novel (novelty condition) open field (Figure 1a). Using ICA, we separated the different hippocampal sources contributing to the LFP [7,31] (Figure 1b). Following this approach, three different components (IC-LFPs) were consistently identified in all subjects. Two of them presented maximum voltage contributions (loadings) in the stratum radiatum (Sch-IC) and stratum lacunosum-moleculare (lm-IC) of CA1 (Figure 1b). These locations matched, respectively, the synaptic terminal fields of the Schaffer collaterals from CA3 and the axons from EC-III (Figure 1c). The third component had its maximum voltage loadings in the mid-molecular layer of the DG (Figure 1b), corresponding with the terminal fields of the perforant pathway (PP-IC) from EC-II to the DG (Figure 1c). Note that the voltage loading of the PP-IC was maximal in the center of the hilar region, although its active current sinks were located in the molecular layer of the DG (Figure 1b). Electric fields generated by both layers of granule cells, above and below the hilus, overlap in this region due to volume conduction, increasing the recorded field potential [38,43].

We analyzed the spectral signature of the different IC-LFPs in familiar vs. novel environments, focusing the analysis on theta and gamma rhythms (Figure 1d). We defined the following frequency bands: theta (6–10 Hz), slow gamma (30–60 Hz), medium gamma (60–100 Hz), and fast gamma (100–150 Hz) [7]. The power spectrum analysis of the IC-LFPs showed in all components a dominant activity in the theta frequency range (~8 Hz) and its first harmonic (~16 Hz; Figure 1d). Theta activity was higher in EC-associated IC-LFPs (lm-IC and PP-IC), increasing in both cases during novelty exploration (Figure 1e; *p* < 0.01, two-way ANOVA between subjects corrected by Holm-Šídák, F(2,12) = 9.25). This result was not related to the running speed of the subjects, as the averaged speed did not differ significantly between conditions (7.2 and 8.1 cm/s in control and novelty, respectively; *p* = 0.26, paired t-test). Although the power spectra did not show defined peaks at higher frequencies, there were significant increases in the slow-gamma and medium-gamma power of the Sch-IC during novelty exploration (Figure 1e; *p* < 0.01, two-way ANOVA corrected by Holm-Šídák, F(2,12) = 10.28 and F(2,12) = 1.232 for slow and medium gamma, respectively). The power increase in particular gamma bands of specific IC-LFPs agrees with previous studies’ reporting that the information reaching CA1 is multiplexed into different gamma bands, with low-gamma associated to the CA3 input and medium-gamma to the EC-III input [10,18,55]. 

### 3.2. Hippocampal Functional Connectivity during Novelty Is Dominated by Theta Inputs from the Entorhinal Cortex

To assess the degree of interactions between IC-LFPs, we performed a PDC analysis (Figure 2). This methodology partializes the connectivity, including all network nodes in the same analysis and differentiating direct from indirect links. Moreover, as it estimates the degree of interaction and directionality at each frequency, facilitates the comparison with the above analysis of power spectrum. The results of the PDC highlighted the importance of the theta rhythm as the main source of directed interactions. In the CA1 region, the connectivity between Schaffer inputs in the stratum radiatum (Sch-IC) and EC-III inputs in the stratum lacunosum-moleculare (lm-IC) was driven by the former, particularly for activities in the theta range (Figure 2a). Interestingly, the influence of Sch-IC over lm-IC decreased in the novelty condition (*p* < 0.05, two-way ANOVA corrected by Holm-Šídák, F(5,24) = 5.689), suggesting a tighter CA3 control over CA1 activity during memory guided exploration in known vs. novel environments (Figure 2a). Activity interactions between Sch-IC and PP-IC were negligible (Figure 2b,e). However, bidirectional interactions between both EC-associated activities (lm-IC and PP-IC) in the theta range were also high (Figure 2c,f), likely reflecting activity interactions between the efferent layers II and III in the upstream entorhinal cortex. Comparing the differences between control and novelty, the PDC revealed an increase in the theta connectivity from PP-IC to lm-IC (*p* < 0.01, two-way ANOVA; Figure 2f), suggesting that the increase in theta power found in both generators (Figure 1e) was likely induced by the higher influence of EC-II over EC-III.

We further analyzed the temporal dynamics of the PDC in those cases demonstrating significant differences between control and novel conditions (Sch-IC → lm-IC and PP-IC → lm-IC; Figure 2g,h). The interaction from the Schaffer component to lm-IC showed higher values at the beginning of the exploration trial in a known environment (control condition) with a slightly decreasing trend over time (R = −0.5; *p* = 0.07). In the novel environment, PDC values were smaller from the beginning of the exploration and remained constant during the trial (R = 0.22; *p* = 0.74). More significant results were found in the PDC from PP-IC to lm-IC, with overall higher values in the novelty condition, and a more pronounced decay over time in both, control and novelty conditions (R = −0.69/−0.73 for control/novelty; *p* < 0.05/0.01). These results complement previous reports on theta synchronization and theta-gamma cross-frequency coupling during the same task showing comparable trends in the dynamics of those metrics [7]. 

Overall, the above results suggest that during novelty exploration, theta activity originating in the EC-II would not only reach the DG, but it would influence CA1 computations by its interaction with EC-III, at the detriment of the contribution from Sch-IC.

### 3.3. Changes in Functional Connectivity Are Based on Linear Interactions

To include the possibility of non-linear interactions between hippocampal regions, we computed a PTE analysis between the three IC-LFPs. PTE measures both linear and nonlinear interactions and, as it is based on information theory, its results can be interpreted in terms of information transfer in the system. Results for control and novel conditions are represented in Figure 3. PTE did not identify any clear unidirectional connectivity, with almost symmetrical values between pairs. The lowest interaction was found between Sch-IC and PP-IC, in good agreement with the linear PDC metric (Figure 2b,e). Information exchange in the system, both inward and outward, was critically dependent on the lm-IC, probably reflecting interactions between theta oscillations across layers as indicated by the PDC (Figure 2a,c,f). The comparison between conditions further confirmed the increase in information transfer from PP-IC to lm-IC during the exploration of a novel environment (Figure 3b; *p* < 0.05, paired t-test between conditions). The only difference found with respect to the PDC analysis was that PTE from Sch-IC to lm-IC yielded no significant differences (*p* = 0.24). 

To complete the analysis, we also characterized the temporal dynamics of the PTE in the same cases of Figure 2g,h (Figure 3c). Similar to the PDC analysis, the connectivity from Sch-IC to lm-IC was higher at the beginning of the task in control, with a linear decay of the connectivity during the trial (R = −0.073, *p* < 0.01). This pair presented the same tendency in the novel environment, but less pronounced (R = −0.48; *p* = 0.06). In the interaction from PP-IC to lm-IC, the PTE followed the same dynamics as the PDC (Figure 2h) in the control condition, presenting the same negative trend (R = −0.68; *p* < 0.01). However, there was no difference between the beginning and the end of the trial in the novel condition, with the PTE value remaining stable during the whole task (R = 0; *p* > 0.5).

Overall, the similarity between PDC and PTE results suggests that the hippocampal functional connectivity is predominately based on linear interactions between theta oscillations (but see Limitations on gamma functional connectivity below).

## 4. Discussion

In this work, we have analyzed the functional connectivity between multiple theta oscillations in the hippocampus. Using ICA, we have disentangled the raw LFPs from the hippocampus into three independent LFP generators (IC-LFPs), one associated with CA3 and two associated with the EC (Figure 1). During the exploration of a novel environment, theta power increased in the generators with an entorhinal origin (Figure 1) and the analysis of functional connectivity using PDC (Figure 2) and PTE (Figure 3) indicated that the change in power was associated with a higher influence of the neuronal activity originated in EC-II (PP-IC) over the one originated in EC-III (lm-IC). Together with previous work revealing multiple theta synchronization states in the hippocampus [7], we hypothesized that the changes in theta synchrony might be driven by EC-II.

### 4.1. Functional Coupling between Schaffer and EC Inputs in CA1

Our results revealed a strong influence of Sch-IC inputs over lm-IC inputs in CA1 (Figure 2). Different neurobiological substrates may explain this interaction. First, as Sch-lC and lm-IC are originated in CA3 and EC-III, respectively, their functional influence could be caused by a common driver to both structures, as the medial septum [56,57,58]. In this scenario, a decrease in the theta PDC from Sch-IC to lm-IC in novelty exploration (Figure 2a) can be interpreted as a shift in the dominance between theta generators. In a known environment, memory guided exploration would be supported by the CA3 Schafer collateral input, theta-paced by the medial septum, dominating over the EC-III theta input in CA1 [18,59]. In a novel environment, exploration relies on external cues, and the multisensory input from the EC would gain relevance, imposing its rhythmicity in the terminal fields of the stratum lacunosum moleculare [60]. A second possibility is that Sch-IC may have an unidirectional connection with lm-IC through the oriens-lacunosum moleculare (OLM) interneurons, which have been suggested as a regulator of the robustness of the theta rhythmicity [61]. The integration of Schaffer and entorhinal inputs in the dendrites of the CA1 pyramidal cells determines their firing and, consequently, the recruitment of OLM interneurons [62]. However, as the OLM cells target the stratum lacunosum-moleculare, they have a direct impact on the transmembrane currents composing the lm-IC generator, but not the Sch-IC [63]. Therefore, a variation in the CA3 output may condition the lm-IC activity through the recruitment of OLM cells, while a variation in the EC-III output would not have the same effect on Sch-IC since OLM terminal fields do not innervate the stratum radiatum. During novelty, the observed increase in the theta power of the lm-IC, but not of the Sch-IC, suggest dominance of the former input over CA1 pyramidal cell firing and OLM cells recruitment, limiting the contribution of the Schaffer inputs and decreasing its causal effect over lm-IC. 

### 4.2. Functional Coupling between Entorhinal Cortical Layers

The intrinsic organization of entorhinal cortical layers is still a matter of debate. In general terms, outputs from CA1 and subiculum arrive at layer V of the EC, which in turn projects to both superficial layers II and III [64,65,66]. Importantly, the existence of connections between neurons in layers II and III has been demonstrated [64,67]. In a previous work, using computational models, we pointed to the relationship between entorhinal layers as a key element to determine the functional connectivity in the hippocampus [30].

In this work, we have found a robust and reciprocal interaction between lm-IC and PP-IC (Figure 2c,f and Figure 3), which likely reflects the interaction between neuronal populations in layers II and III. As discussed for Sch-IC and lm-IC, this interaction could be materialized in a local circuit, or driven by a common input to both EC layers. One possibility for the local interaction could be through chandelier interneurons, which have been found in EC-II and EC-III and have axon terminals innervating both layers [67]. Interestingly, while principal cells in both EC layers fire in antiphase relative to the theta rhythm recorded in CA1, interneurons fire in phase [68]. One explanation for the increased effect of PP-IC over lm-IC during novelty exploration could be that interneurons in the EC set the phase of the hippocampal theta oscillations by determining the timing of principal cells firing in layers II and III, respectively [7]. We hypothesize that the increased functional connectivity reflects a modulation of EC-III principal cells by EC-II interneurons (e.g., chandelier interneurons), which, recruited by EC-II principal cells, would set the phase of the EC-III output, synchronizing both theta oscillations in the hippocampus. 

### 4.3. Functional Role of the Entorhinal Cortex in Novelty and Navigation

What could be the implications of EC-II as a driver of theta synchronization in the hippocampal formation? We theorized that the increased activity in PP-IC during novelty reflects a shift in the navigation mechanism. In a known environment, a cognitive map, based on place cells, would be optimal to recognize the current location and find the best pathway to a target over all the previously visited regions [69]. However, this mechanism suffers when traversing novel areas. Grid cells in the EC, on the contrary, fire in regular spatial patterns, filling the whole region [70]. Together with head-direction cells, also found in the EC, grid cells would allow the prediction of future locations directly ahead of the animal’s nose (linear look-ahead) [71,72]. Since grid cells predominate in EC-II, while conjunctive cells, i.e., cells with a combination of grid and head-direction properties [73], are located primarily in EC-III, grid to head-direction cells communication should translate into a measurable directed functional connectivity from EC-II to EC-III, as we find in our study using LFP current generators.

Taken together, when introducing a new stimulus (novelty condition), the spatial navigation would rely on the grid cells and, therefore, EC-II would drive the information flow to the hippocampus and EC-III.

### 4.4. Limitations on Gamma Functional Connectivity

Some characteristics of the used methodology limit the identification of possible interactions between gamma oscillations in the hippocampus. First, gamma activity tends to occur in discrete bursts at specific phases of the theta rhythm [7,10,41]. Thus, being a non-stationary process, the identification of gamma interactions by PDC and PTE analysis is not optimal [74]. Second, gamma oscillations are multiplexed in the hippocampus into several frequency bands [7,10,16]. Linear methods such as PDC only measure interactions at the same frequency and, therefore, interactions between different gamma frequency bands might exist, but they remain undetected by the present methods. Third, the signal-to-noise ratio of brain signals decreases with frequency [75], with theta and gamma showing powers with different orders of magnitude (Figure 1e). When computing the functional connectivity in the time domain, the interactions between high power theta components will predominate. Therefore, while PTE methods are not limited by linear constraints, they measure the global connectivity, that may be mainly explained by the theta oscillations.

Finally, metrics such as PDC and PTE provide information about direct connections based on the ability to “partialize” the connectivity in a reduced model. Therefore, they require temporal information of all nodes in the network to infer the directionality. In our analysis, three generators were included in the analysis (Sch-IC, lm-IC, and PP-IC), but additional connections mediated by other regions cannot be discarded.

## 5. Conclusions

We have investigated the functional interactions in the hippocampus between pathway-specific LFP generators and demonstrated an increase in the interactions between the EC-associated generators during novelty exploration, consistent with an increased influence of activity originated in EC-II over EC-III in the theta range, and a decreased influence in CA1 of the Schaffer generator over the EC-III input. We hypothesize that the increased theta power and synchrony, measured in the hippocampus during novelty tasks [7], is driven by EC-II activity.

## Figures and Tables

**Figure 1 biology-10-00692-f001:**
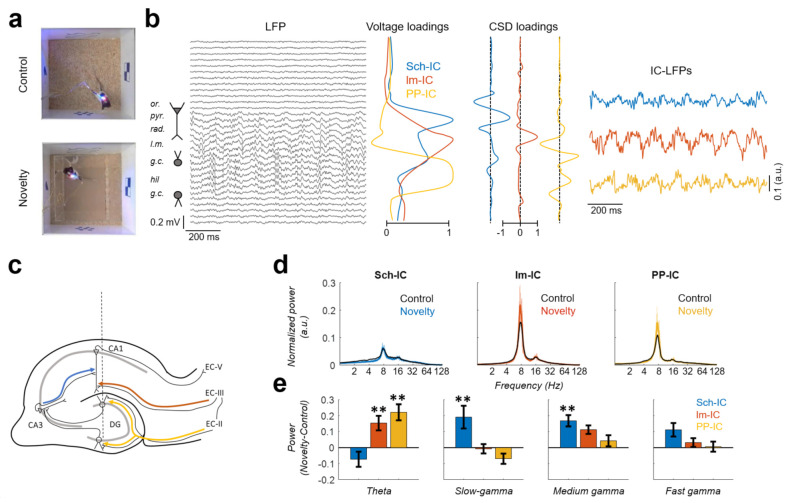
Activity of the pathway-specific LFPs during familiar vs. novelty explorations. (**a**) Example of one video frame acquired during the exploration of familiar (top) and novel (bottom) environments. (**b**) Decomposition of the hippocampal LFPs in their sources using ICA. Left: LFP traces recorded along the dorsal hippocampus (or, stratum oriens; pyr, pyramidal layer; rad, stratum radiatum; lm, stratum lacunosum-moleculare; gc, granule cell layer; hil, hilus.). Middle: Voltage and current source density (CSD) loadings of the independent components (IC) of the LFP (IC-LFPs), which represent the contribution of the sources to each LFP channel. Note that one LFP channel may be contributed by several sources. Right: IC-LFPs obtained for the represented LFPs. (**c**) Scheme of the main hippocampal regions. The dashed line represents the location of the electrode. Color lines identify the pathways associated to the IC-LFPs: Schaffer collateral in blue, temporoammonic pathway (EC-III efferences to CA1) in red and perforant pathway (EC-II efferences to DG) in yellow. (**d**) Averaged power spectrum across subjects (*n* = 5) for each signal in control (black) and novelty (color; mean ± s.e.m.). (**e**) Difference between the power in novelty versus control at different frequency bands (mean across subjects ± s.e.m.; ** *p* < 0.01, *n* = 5).

**Figure 2 biology-10-00692-f002:**
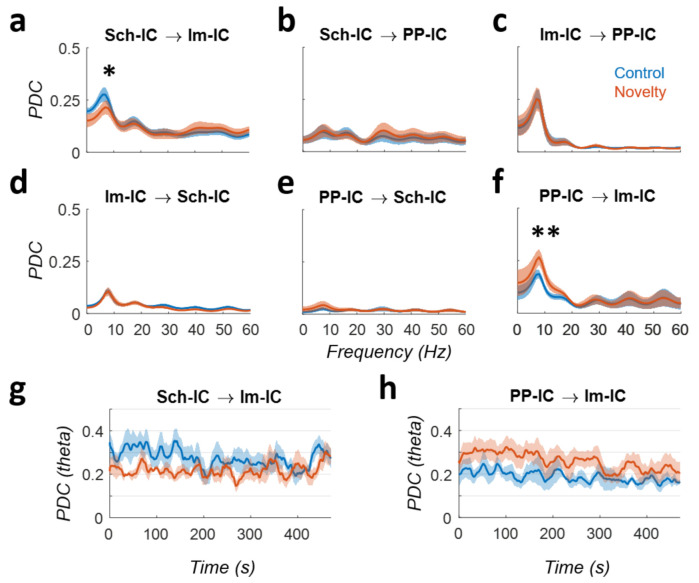
Directed functional connectivity. (**a**–**f**): PDC analysis between IC-LFPs in control (blue) vs. novelty (red) conditions for each pair of IC-LFPs and in both directions (mean across subjects ± s.e.m.; * *p* < 0.05, ** *p* < 0.01, *n* = 5). Arrows indicate the directionality of the interaction. (**g**,**h**): Time evolution of the PDC at theta frequency during the tasks for the two pairs of IC-LFPs showing significant differences in panels (**a**,**h**).

**Figure 3 biology-10-00692-f003:**
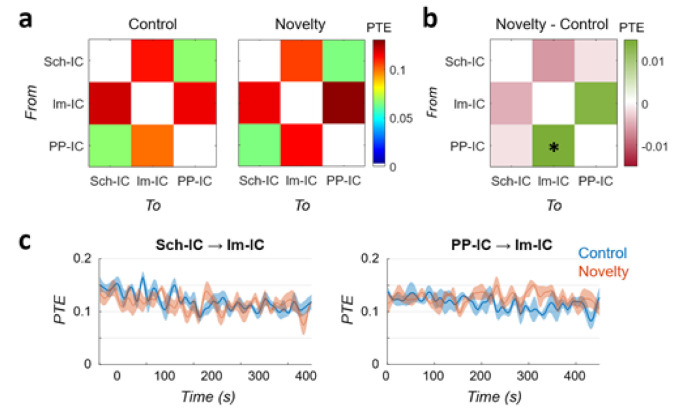
Functional connectivity in the hippocampus based on Information Theory. (**a**) PTE analysis between IC-LFPs during the control and novelty session (mean value across subjects, *n* = 5). Each value represents the connectivity from the sender in the *y*-axis to the receiver indicated in the *x*-axis. (**b**) Difference between PTE matrices in novelty minus control data (* *p* < 0.05). (**c**) Time evolution of the PTE during the tasks for the two pairs of IC-LFPs showing significant differences in PDC and/or PTE.

## Data Availability

All the data used in this work are available at http://dx.doi.org/10.20350/digitalCSIC/12537 (accessed on 19 July 2021).

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
