# Peer review of "Functional Interactions between Entorhinal Cortical Pathways Modulate Theta Activity in the Hippocampus"

_biology, 2021, doi:10.3390/biology10080692_

Round 1

Reviewer 1 Report

In general, this current work provided a new perspective of hippocampal theta activity and synchrony during novelty exploration, which closely link increased theta oscillation to an increase in the information flow from EC-II to EC-III through statistical methods Partial Directed Coherence and Partial Transfer Entropy. It is an interesting work and scientifically adequate for publishing in Biology after some major revisions.

Major:

  1. In Figure 1 there are multiple key information missing. In panel b, there is no scale bar in y-axis for LFP traces (left), no x-y axis for IC-LFPs and represented LFPs (middle and right), which is hard to depict the quality of recorded response. In panel d, there is no clear, direct description about sample number for the averaged results.
  2. For figure 2-3 one case similar as mentioned above, the sample number should be specified in figure legend or mentioned in the main content clearly, directly for grouped data, such as recorded unit/recording number, animal number etc.;
  3. The middle trace in Figure 1b, voltage loading of the IC-LFPs, shared very high similarity with Figure 1b in ref 7 (López-Madrona, V.J.; Pérez-Montoyo, E.; Á lvarez-Salvado, E.; Moratal, D.; Herreras, O.; Pereda, E.; Mirasso, C.R.; Canals, S. Different Theta Frameworks Coexist in the Rat Hippocampus and Are Coordinated during Memory Guided and Novelty Tasks. eLife 2020, 9, doi:10.7554/eLife.57313.);
  4. In Line 111-113, ‘Five adult male Long-Evans rats (250-300 g) were implanted with a 32 channels silicon probe across the dorsal hippocampus. Details of the surgery can be found elsewhere [7].’ However, it is too simplified and limited information provided here. As this is another complete work, and some fundamental content, such as manufacture information of the probe and spectral/temporal roundness of picking up signals by 32 channels silicon probe, the recording coordination and depth (whether it covers the ‘theta generators’ in hippocampus region, referring to areas authors are majorly concerned), and post hoc histological method used to confirm CA3 (Schaffer collaterals), the entorhinal cortex layers II (EC-II, performant pathway) and III (EC-III, temporoammonic pathway) should be at least briefly describe. As these material and method are both fundamental and critical to evaluate the soundness of this work.
  5. As authors mentioned in ‘Data acquisition’ and ‘Independent component analysis’ parts, ref. 7 is the one basal proof, and these two works do focus on a similar topic. However, in ref. 7 Figure 6b, the Inter-cycle phase clustering between theta oscillations of animal subjects during the novel versus control environment exploration show dynamic temporal attributes. In this work, related to Figure 1-3:

         â‘  how long IC-LFPs are sampled from? The full length of 10 mins as indicated in line 117-122?

        â‘¡ whether the PDC analysis and PTE analysis (Figure 2-3) of IC-LFPs during the control and novelty session also exhibit such  temporal dynamic in parallel during mismatch novelty? This could be very interesting to inspect as indicated in ref. 7.

  1. As authors mentioned in Line 220-222, ‘This result was not related to the running speed of the subjects, as the averaged speed did not differ significantly between conditions (26 and 29 cm/s in control and novelty, respectively; p=0.26, paired t-test).’, but in ref. 27 which resembles to this study a lot, in Figure 6c, the movement velocity of the animals during control and novelty are way below 26 or 29 cm/s (if averaged t1 to t3 the mean speed is below 10 cm/s). Why the activity level could be so different among rats in these two works? Considering same behavior paradigm and close animal number here. Authors also need to clarity that variables such us running speed and time are involving marginally in the final conclusions here.

Minor:

  1. In Line 178, ‘eMVAR framework’, authors need to state its full name before abbreviations;
  2. Figure 1c, suggesting to use the example slice image indicating the position of the recording electrode in one experiment animal used in this study, instead if a simplified diagram;
  3. The symbols → in Figure 2 could be explained in its figure caption below;
  4. There is a typo in line 267, ‘*/** p<0.05/0.01’, if it is not put it on purpose;
  5. There are some necessary citations missing, such as, in Line 308-313, ‘In a known environment, memory guided exploration would be supported by the CA3 Schafer collateral input, theta-paced by the medial septum, dominating over the EC theta input in the temporoammonic pathway. In a novel environment, exploration relies on external cues, and the multisensory input from the EC would gain relevance, imposing its rhythmicity in the terminal fields of the stratum lacunosum moleculare.’, and, in Line 318-320, ‘However, as the OLM cells target the stratum lacunosum-moleculare, they have a direct impact on the transmembrane currents composing the lm-IC generator, but not the Sch-IC.’.
  6. Why the slow gramma band in Figure 2 is only shown to 50 Hz, as authors defined here as slow gamma (30–60 Hz). Is only incomplete slow gamma band displayed in Figure 2? Is it because what authors discussed in section 4.4?
  7. In Figure 1, panel d is missing in the figure;
  8. In section 3.2, please consider add extra subpanels in Figure 2 and cite Figure 2 result respectively in the text. In current form, it could be bit lost for general readers.

Author Response

In general, this current work provided a new perspective of hippocampal theta activity and synchrony during novelty exploration, which closely link increased theta oscillation to an increase in the information flow from EC-II to EC-III through statistical methods Partial Directed Coherence and Partial Transfer Entropy. It is an interesting work and scientifically adequate for publishing in Biology after some major revisions.

We thank the reviewer for her/his positive appreciation of our work.

Major:

1. In Figure 1 there are multiple key information missing. In panel b, there is no scale bar in y-axis for LFP traces (left), no x-y axis for IC-LFPs and represented LFPs (middle and right), which is hard to depict the quality of recorded response. In panel d, there is no clear, direct description about sample number for the averaged results.

We apologize for the missing information. All requested details have been included in the revised version.

2. For figure 2-3 one case similar as mentioned above, the sample number should be specified in figure legend or mentioned in the main content clearly, directly for grouped data, such as recorded unit/recording number, animal number etc.;

We have indicated sample sizes in all figures and additional information, as requested.

3. The middle trace in Figure 1b, voltage loading of the IC-LFPs, shared very high similarity with Figure 1b in ref 7 (López-Madrona, V.J.; Pérez-Montoyo, E.; Álvarez-Salvado, E.; Moratal, D.; Herreras, O.; Pereda, E.; Mirasso, C.R.; Canals, S. Different Theta Frameworks Coexist in the Rat Hippocampus and Are Coordinated during Memory Guided and Novelty Tasks. eLife 2020, 9, doi:10.7554/eLife.57313.);

We have replaced LFP traces and the corresponding IC-LFPs, time courses and loadings, in Figure 1b. We have used the recordings from a different subject, to make the differences more evident.

4. In Line 111-113, ‘Five adult male Long-Evans rats (250-300 g) were implanted with a 32 channels silicon probe across the dorsal hippocampus. Details of the surgery can be found elsewhere [7].’ However, it is too simplified and limited information provided here. As this is another complete work, and some fundamental content, such as manufacture information of the probe and spectral/temporal roundness of picking up signals by 32 channels silicon probe, the recording coordination and depth (whether it covers the ‘theta generators’ in hippocampus region, referring to areas authors are majorly concerned), and post hoc histological method used to confirm CA3 (Schaffer collaterals), the entorhinal cortex layers II (EC-II, performant pathway) and III (EC-III, temporoammonic pathway) should be at least briefly describe. As these material and method are both fundamental and critical to evaluate the soundness of this work.

The reviewer is right, we simplified the description of the experimental methods too much. We have now updated the methods including a new section “2.1 Animals and surgery” to provide more details (see below).

2.1. Animals and surgery

Five adult male Long-Evans rats (250-300 g) were each implanted with a 32 chan-nels silicon probe (Neuronexus Technologies, Michigan, USA) across the dorsal hippocampus (data are available at http://dx.doi.org/10.20350/digitalCSIC/12537). Data from the same subjects have been used in a previous study [7]. An Ag/AgCl wire (World Precision Instruments, Florida, USA) electrode was placed in contact with the skin on the sides of the surgery area and used as ground. The data were acquired at 5 kHz, with an analog high-pass filter at 0.5 Hz. After digitalization, we low-pass fil-tered the signals at 300 Hz, removed the line noise at 50 Hz and its first harmonic with Notch filters and down-sampled the signals at 2.5 kHz. At the end of the experiments, animals were perfused with 4% paraformaldehyde, and the final position of the electrodes confirmed histologically.

The rats were left for at least 10 days after the surgery, until they recovered complete-ly. During the first 72 hr, animals were injected subcutaneously with analgesic twice per day (Buprenorphine, dose 2–5 μg/kg, RB Pharmaceutical Ltd., Berkshire, UK). An-tibiotic (Enrofloxacin, 10 mg/kg, Syva, León, Spain) dissolved in the drinking water was also provided during the first post-surgery week. Behavioral training did not start until the animals showed no signs of discomfort with the manipulation of the im-plants.

5. As authors mentioned in ‘Data acquisition’ and ‘Independent component analysis’ parts, ref. 7 is the one basal proof, and these two works do focus on a similar topic. However, in ref. 7 Figure 6b, the Inter-cycle phase clustering between theta oscillations of animal subjects during the novel versus control environment exploration show dynamic temporal attributes. In this work, related to Figure 1-3:

How long IC-LFPs are sampled from? The full length of 10 mins as indicated in line 117-122?

Whether the PDC analysis and PTE analysis (Figure 2-3) of IC-LFPs during the control and novelty session also exhibit such  temporal dynamic in parallel during mismatch novelty? This could be very interesting to inspect as indicated in ref. 7.

We thank the reviewer for rising this important point and suggestion. In this work, we analyzed and averaged the whole temporal window, as indicated in the text. However, following the reviewer’s suggestion, we have included a new analysis of the temporal dynamics of the PDC and PTE for those two pairs of IC-LFPs showing significant differences between control and novelty (Sch-IC ? lm-IC and PP-IC ? lm-IC). We have incorporated this new result in the text and Figures 2 and 3, as follows:

We further analyzed the temporal dynamics of the PDC in those cases demonstrating significant differences between control and novel conditions (Sch-IC à lm-IC and PP-IC à lm-IC; Figures 2g and h). The interaction from the Schaffer component to lm-IC showed higher values at the beginning of the exploration trial in a known environment (control condition) with a slightly decreasing trend over time (R = -0.5; p=0.07). In the novel environment, PDC values were smaller from the beginning of the exploration and remained constant during the trial (R = 0.22; p=0.74). More significant results were found in the PDC from PP-IC to lm-IC, with overall higher values in the novelty condition, and a more pronounced decay over time in both, control and novelty conditions (R = -0.69/-0.73 for control/novelty; p<0.05/0.01). These results complement previous reports on theta synchronization and theta-gamma cross-frequency coupling during the same task showing comparable trends in the dynamics of those metrics [7].

To complete the analysis, we also characterized the temporal dynamics of the PTE in the same cases of Figure 2g and h (Figure 3c). Similar to the PDC analysis, the connectivity from Sch-IC to lm-IC was higher at the beginning of the task in control, with a linear decay of the connectivity during the trial (R=-0.073, p<0.01). This pair presented the same tendency in the novel environment, but less pronounced (R=-0.48; p=0.06). In the interaction from PP-IC to lm-IC, the PTE followed the same dynamics as the PDC (Figure 2h) in the control condition, presenting the same negative trend (R=-0.68; p<0.01). However, there was no difference between the beginning and the end of the trial in the novel condition, with the PTE value remaining stable during the whole task (R=0; p>0.5).

6. As authors mentioned in Line 220-222, ‘This result was not related to the running speed of the subjects, as the averaged speed did not differ significantly between conditions (26 and 29 cm/s in control and novelty, respectively; p=0.26, paired t-test).’, but in ref. 27 which resembles to this study a lot, in Figure 6c, the movement velocity of the animals during control and novelty are way below 26 or 29 cm/s (if averaged t1 to t3 the mean speed is below 10 cm/s). Why the activity level could be so different among rats in these two works? Considering same behavior paradigm and close animal number here. Authors also need to clarity that variables such us running speed and time are involving marginally in the final conclusions here.

The reviewer is absolutely right. The running speed values reported in the original manuscript were not in the units indicated in the text (cm/s). This mistake explains the differences noted by the reviewer. We apologize for this error, and have corrected now the values in the revised manuscript (7.2 and 8.1 cm/s for control and novelty, respectively).

Minor:

1. In Line 178, ‘eMVAR framework’, authors need to state its full name before abbreviations;

Full name included now.

2. Figure 1c, suggesting to use the example slice image indicating the position of the recording electrode in one experiment animal used in this study, instead if a simplified diagram;

As the data used in this study is the same as in our previous study López-Madrona et al. 2020, where details and images with the electrode position were given, we think it would be better not to repeat the same information in a main figure.

3. The symbols → in Figure 2 could be explained in its figure caption below;

We have explained the arrows as follows: “Arrows indicate the directionality of the interaction”

4. There is a typo in line 267, ‘*/** p<0.05/0.01’, if it is not put it on purpose;

We have now separated both statistical thresholds as: *p<0.05, **p<0.01, n=5

5. There are some necessary citations missing, such as, in Line 308-313, ‘In a known environment, memory guided exploration would be supported by the CA3 Schafer collateral input, theta-paced by the medial septum, dominating over the EC theta input in the temporoammonic pathway. In a novel environment, exploration relies on external cues, and the multisensory input from the EC would gain relevance, imposing its rhythmicity in the terminal fields of the stratum lacunosum moleculare.’, and, in Line 318-320, ‘However, as the OLM cells target the stratum lacunosum-moleculare, they have a direct impact on the transmembrane currents composing the lm-IC generator, but not the Sch-IC.’.

We have included new references supporting these statements.

6. Why the slow gramma band in Figure 2 is only shown to 50 Hz, as authors defined here as slow gamma (30–60 Hz). Is only incomplete slow gamma band displayed in Figure 2? Is it because what authors discussed in section 4.4?

Thanks for this highlighting this mismatch in the representation. In the revised manuscript, in the new Figure 2, we have now extended the x axis up to 60 Hz to fully cover the slow-gamma frequency band.

7. In Figure 1, panel d is missing in the figure;

Figure 1 has been reviewed.

8. In section 3.2, please consider add extra subpanels in Figure 2 and cite Figure 2 result respectively in the text. In current form, it could be bit lost for general readers.

We have corrected all the issues addressed by the reviewer and the typos found in the text.

Reviewer 2 Report

Functional interactions between entorhinal cortical pathways 2 modulate theta activity in the hippocampus 3

Víctor J. López-Madrona 1,* and Santiago Canals 2,*

REVIEW

This manuscript discusses the important issue of mechanisms within the hippocampus involved in novelty processing at the level of anatomical pathways and physiological activity, notably theta activity. The study is very well thought out and designed, including direct comparisons of novelty and no-novelty conditions, with clear data analysis of electrophysiological field potentials and the generators of them. It is very well organized and written. In sum, an excellent piece of work.

The central message is that the prominent “theta” activity (6-10 Hz) observed in hippocampus derives from separate generators in different circuits, and they may be segregating or multiplexing different flows of information. If so, and if the independent generators can be identified, it should be possible to look at the sequential flow of information. Granger ‘casuality’ (which is a really a kind of quasi-causality) is used to examine what comes first and what second for target regions receiving inputs from two regions.  On this basis, the two central claims of the manuscript are that (a) input from EC-layer II via the perforant path dominates during novelty, but (b) the Schaffer collateral input to CA1 dominates over the EC-layer III temporamonnic pathway (at least in determining theta activity). This is a novel result worth publishing.

GENERAL COMMENT

However, given the complexity of the connections and pathways referred to in the text, the authors should clarify their terms from the outset and be consistent in using them to keep the reader, even expert readers, on track.

A SUGGESTION

Perhaps, perforant (perhaps  shortened to ECII-DG) and temporo-ammonic (perhaps  shortened to ECIII-CA1) pathways may work well as terms, even if the authors add subtleties to them as required in context. 

In fact, the terms temporo-ammonic and perforant pathway do appear late in the text (lines 209, and 239-Fig. 1 d, respectively). They seem both very pertinent here and could be used as conceptual anchor points for ECII-DG etc.

SPECIFIC COMMENTS

Do really EC layer V neurons project DIRECTLY back to CA1? Or do they do it via Layer II or III of EC, as the authors refer in 61,62?? That is the sort of dogma (also recently shown in detail by M Witter’s group (see for example Ohara et al., 2018 Cell Reports 24, 107–116).  The paragraph dedicated to this issue in the DISCUSSION in 4.2 section does not address/responds to this issue directly.

ABSTRACT

Lines 32- 33:

“We found a significant unidirectional interaction from the Schaffer component over the EC-III component in CA1, and bidirectional interactions between the two EC components”.

This phrase is confusing for two reasons:

The term bidirectional interactions between the two components is unclear. Do the authors mean ECIII and ECII or ECIII-DG, in whatever case, authors should be precise, also because the unidirectional nature of the hippocampal intrinsic connections.

METHODS

Sections 2.3  and 2.4 are very difficult to  follow are very  difficult  to read unless you are expert in the field, which may  not be the  case of  the readers of this journal.

FIGURES

Abbreviations should be complete appear at the end of the figure legend in alphabetical order.  IC is missing from the legend in Fig. 1. In fact, IC seems to refer to the LFP component, but this should be made more explicit (in the abbreviations, and the text), otherwise, it seems to refer to some additional node of the pathway. In fact, it is everywhere, so it could be easlily omitted as it does not add anything, and yet it confuses.

This also applies to the first paragraph of the discussion.

DISCUSSION

In contrast with the authors first conclusion/hypothesis, it seems, by looking at figures 1 and 2, that the source data analysis indicates it is EC-Layer III after all the one driving the novelty processing in the hippocampus. This should be clarified.

Authors should consult the human hippocampus high power Tesla (9) fMRI work, as the opposite hypothesis has been proposed there. See the work of E Duzel (Maass et al., 2014, Nature Communications volume 5, Article number: 5547).

TYPOS

Line 104: Shouldn’t it be “separate”

Line 111: were each implanted

Line 115: methacrylate is OK, but the usual term is “plexiglass” (in the US)

Line 129: this defines ICA and the acronym only is used on line 138. That’s fine.  But then line 146 uses the acronym IC-LFP which could be “intracranial local field potential” or more likely “independent component local field potential”.  The context emerging from line 142-44 favors the former interpretation, but I actually suspect it is the latter based on the use of a term like Sch-IC in Figure 1. This is an example of the “death by acronym” issue pervading the manuscript as the letters I and C may mean different things across only a few sentences – or they may not. A read through with caution on behalf of the uninitiated should be sufficient to solve this kind of problem.  Note general point above and possibility of using the term temporo-ammonic additionally on line 150.

Author Response

This manuscript discusses the important issue of mechanisms within the hippocampus involved in novelty processing at the level of anatomical pathways and physiological activity, notably theta activity. The study is very well thought out and designed, including direct comparisons of novelty and no-novelty conditions, with clear data analysis of electrophysiological field potentials and the generators of them. It is very well organized and written. In sum, an excellent piece of work.

We thank the reviewer for her/his very positive appreciation of our work.

The central message is that the prominent “theta” activity (6-10 Hz) observed in hippocampus derives from separate generators in different circuits, and they may be segregating or multiplexing different flows of information. If so, and if the independent generators can be identified, it should be possible to look at the sequential flow of information. Granger ‘casuality’ (which is a really a kind of quasi-causality) is used to examine what comes first and what second for target regions receiving inputs from two regions.  On this basis, the two central claims of the manuscript are that (a) input from EC-layer II via the perforant path dominates during novelty, but (b) the Schaffer collateral input to CA1 dominates over the EC-layer III temporamonnic pathway (at least in determining theta activity). This is a novel result worth publishing.

GENERAL COMMENT

However, given the complexity of the connections and pathways referred to in the text, the authors should clarify their terms from the outset and be consistent in using them to keep the reader, even expert readers, on track.

We appreciate this point raised by the reviewer. Although we have used different names for the same pathway in an attempt to avoid excessive repetition of the same terms, we understand that this can be confusing to the general reader. Therefore, we have revised the text to clarify terminology and eliminate excessive use of several terms for the same pathway.

A SUGGESTION

Perhaps, perforant (perhaps  shortened to ECII-DG) and temporo-ammonic (perhaps  shortened to ECIII-CA1) pathways may work well as terms, even if the authors add subtleties to them as required in context. 

In fact, the terms temporo-ammonic and perforant pathway do appear late in the text (lines 209, and 239-Fig. 1 d, respectively). They seem both very pertinent here and could be used as conceptual anchor points for ECII-DG etc.

As indicated in the previous general comment, we have updated the text clarifying the pathways. To keep the same terminology as in our previous work (López-Madrona et al., 2020), we would like to maintain the terms Sch-IC, lm-IC and PP-IC in the text. Nevertheless, following the reviewer’s advice, we have minimized the use of temporo-ammonic pathway (replaced or clarified as the input from EC-III to CA1) and we have remarked along the text that the perforant pathway refers to the EC-II input to DG. 

SPECIFIC COMMENTS

Do really EC layer V neurons project DIRECTLY back to CA1? Or do they do it via Layer II or III of EC, as the authors refer in 61,62?? That is the sort of dogma (also recently shown in detail by M Witter’s group (see for example Ohara et al., 2018 Cell Reports 24, 107–116).  The paragraph dedicated to this issue in the DISCUSSION in 4.2 section does not address/responds to this issue directly.

In the manuscript, we discussed the main communication pathway from the hippocampus to the EC, which is represented by the projection from CA1 and subiculum to the EC layer V. In this regard, we stick to the dominant view in the field, as indicated by the reviewer. We are not aware of a backward projection from EC-V to CA1. However, in case this projection would exist, it would not have a strong impact on the results reported in this work, as we focus on the interaction between the inputs from superficial layers II and III of the EC to the hippocampus.

We have included the reference suggested by the reviewer (Ohara et al. 2018) in this section to support the discussion of the entorhinal-hippocampal loop.

ABSTRACT

Lines 32- 33:

“We found a significant unidirectional interaction from the Schaffer component over the EC-III component in CA1, and bidirectional interactions between the two EC components”.

This phrase is confusing for two reasons:

The term bidirectional interactions between the two components is unclear. Do the authors mean ECIII and ECII or ECIII-DG, in whatever case, authors should be precise, also because the unidirectional nature of the hippocampal intrinsic connections.

Thanks for highlighting this point. We have rephrased the sentence as follows: “We found a significant directed interaction from the Schaffer input over the EC-III input in CA1, and a bidirectional interaction between the inputs in the hippocampus originating in the EC, likely reflecting the connection between layers II and III.”

METHODS

Sections 2.3 and 2.4 are very difficult to follow are very difficult to read unless you are expert in the field, which may not be the case of the readers of this journal.

In the revised manuscript we have explained in more detail the directed functional connectivity methods.

FIGURES

Abbreviations should be complete appear at the end of the figure legend in alphabetical order.  IC is missing from the legend in Fig. 1. In fact, IC seems to refer to the LFP component, but this should be made more explicit (in the abbreviations, and the text), otherwise, it seems to refer to some additional node of the pathway. In fact, it is everywhere, so it could be easlily omitted as it does not add anything, and yet it confuses.

This also applies to the first paragraph of the discussion.

As indicated above, we would like to keep the terminology of the independent components (Sch-IC, lm-IC, PP-IC and IC-LFPs) to maintain the coherence with our previous work (López-Madrona et al., 2020). However, we understand the reviewer’s concerns and we have clarified the terms when used to avoid misunderstandings.

DISCUSSION

In contrast with the authors first conclusion/hypothesis, it seems, by looking at figures 1 and 2, that the source data analysis indicates it is EC-Layer III after all the one driving the novelty processing in the hippocampus. This should be clarified.

As we show in the results section and highlight in the discussion, during the exploration of a novel environment, theta power increased only in the generators with an entorhinal origin (Figure 1e, first histogram showing significant increases in theta power in novelty vs. control for the EC-II originated activity in yellow, and the EC-III originated activity in orange). Changes in gamma activity in novelty vs. control were restricted to the Schaffer originated activity (Sch-IC). Furthermore, the functional connectivity analysis indicated that EC-II activity was driving the change in theta activity, as PDC in the theta range from EC-II to EC-III was increased during novelty (Figure 2f). This result was confirmed by the PTE analysis (Figure 3, only significant difference found from EC-II to ECIII direction). Changes in functional interactions, as measured by PDC or PTE, were not found for any gamma band in any of the LFP generators. Directed influences from EC-III activity over EC-II or Schaffer activities were not found in any frequency band. Overall, we interpreted that the increase in theta power found during novelty indicated that the change in power was associated with a higher influence of the neuronal activity originated in EC-II (PP-IC) over the one originated in EC-III (lm-IC).

We hope this further clarifies our interpretation of the results.

Authors should consult the human hippocampus high power Tesla (9) fMRI work, as the opposite hypothesis has been proposed there. See the work of E Duzel (Maass et al., 2014, Nature Communications volume 5, Article number: 5547).

We thank the reviewer for bringing this work to our attention. It is an interesting work in humans closely related to the topic of our research. However, some important limitations of the fMRI technique preclude a direct comparison. Besides the different nature of the signals under study, hemodynamic vs. neurophysiological, and the yet unsolved important issue of the potentially different neurovascular coupling mechanisms in different cortical layers (Goense, J., Merkle, H. & Logothetis, N. K. High-resolution fMRI reveals laminar differences in neurovascular coupling between positive and negative BOLD responses. Neuron 76, 629–639. 2012), the most important limitation here is the spatial resolution.

As the authors state in their study “… we divided the EC into three equally sized subregions: superficial, middle and deep. Although the anatomical layering of the EC could not be identified on our MR images, previous ultra-high-resolution ex vivo MRI studies (13,14) suggest that our superficial EC subregion very likely corresponds to EC input layers (covering layer II and probably parts of layer III), whereas our deep EC subregion likely covers the output layers of the EC (V and VI).” This means that the authors obtain 1 fMRI (BOLD) signal combining EC-II and EC-II. Therefore, the main focus of our work, the EC-II to EC-III interaction, couldn’t be considered by the authors. The authors were concern with the output of CA1 to deep EC (a combinations of EC-V + EC-VI in their fMRI study) and the input from EC superficial layers (ECII+ECIII) to DG/CA3.

TYPOS

Line 104: Shouldn’t it be “separate”.

Line 111: were each implanted.

Line 115: methacrylate is OK, but the usual term is “plexiglass” (in the US).

We have corrected all the typos highlighted by the reviewer.

Line 129: this defines ICA and the acronym only is used on line 138. That’s fine.  But then line 146 uses the acronym IC-LFP which could be “intracranial local field potential” or more likely “independent component local field potential”.  The context emerging from line 142-44 favors the former interpretation, but I actually suspect it is the latter based on the use of a term like Sch-IC in Figure 1. This is an example of the “death by acronym” issue pervading the manuscript as the letters I and C may mean different things across only a few sentences – or they may not. A read through with caution on behalf of the uninitiated should be sufficient to solve this kind of problem.  Note general point above and possibility of using the term temporo-ammonic additionally on line 150.

This question has been addressed above (FIGURES question).

Reviewer 3 Report

The theta oscillations in the hippocampus organize neuronal firing during context exploration and memory formation. López-Madrona and Canals have analyzed the connectivity between theta generators during the novelty exploration. The authors found that during novelty exploration, the theta and gamma oscillations are enhanced, hippocampal functional connectivity is dominated by theta inputs from EC, and changes in the functional connectivity are based on linear interactions. The manuscript is well written and clearly presented. I only have one suggestion. Could the authors discuss what could potentially go wrong with their data analysis methods, ICA, PDC and PTE?

Author Response

We appreciate the positive comments of the reviewer and, following her/his advice, we have updated the methods and discussion including considerations and limitations for ICA, PDC and PTE.

ICA (methods):

A main difficulty when computing ICA is the correct identification of physiological sources. As ICA may identify as many possible sources as the number of LFP signals, additional constraints are necessary to ensure the origin, either neuronal or noisy, of each generator. First, the anatomical distribution of each current generator is assumed to be fixed, thus the voltage distribution or topography of the generator should be stable over time. This can be tested by fragmenting the signal into smaller time windows and computing ICA on each fragment separately. Moreover, a certain degree of similarity between generators is expected in different subjects with similar electrode implantations. Second, the topography should follow the specific distribution of axons and dendrites composing the source. This requires a previous knowledge of the anatomical substrate and realistic computational models to simulate the currents. Third, to ensure the synaptic specificity of each generator, the different substructures can be independently modulated, for example with pharmacology, electrical or optogenetic stimulation, confirming that only the associated generator is affected by the manipulation.

PDC and PTE (discussion):

Finally, metrics as PDC and PTE provide information about direct connections based on the ability to “partialize” the connectivity in a reduced model. Therefore, they require temporal information of all nodes in the network to infer the directionality. In our analysis, three generators were included in the analysis (Sch-IC, lm-IC and PP-IC), but additional connections mediated by other regions cannot be discarded.

Round 2

Reviewer 1 Report

It is nicely presented in the revised version, and all my concerns has been addressed and provided with additional proof.

Author Response

We thank the reviewer for her/his positive appreciation of our work.